# Effects of *Trichoderma atroviride* SG3403 and *Bacillus subtilis* 22 on the Biocontrol of Wheat Head Blight

**DOI:** 10.3390/jof8121250

**Published:** 2022-11-27

**Authors:** Hongyi Liu, Tingting Li, Yaqian Li, Xinhua Wang, Jie Chen

**Affiliations:** 1School of Agriculture and Biology, Shanghai Jiao Tong University, Shanghai 200240, China; 2State Key Laboratory of Microbial Metabolism, Shanghai Jiao Tong University, Shanghai 200240, China

**Keywords:** *Bacillus subtilis*, co-culture, *Fusarium graminearum*, *Trichoderma atroviride*, toxins

## Abstract

Wheat head blight caused by *Fusarium graminearum* is one of the major wheat diseases in the world; therefore, it is very significant to develop an effective and environmentally friendly microbial fungicide against it. *Trichoderma atroviride* and *Bacillus subtilis* are widely applied biocontrol microorganisms with separate advantages; however, little work has been conducted for synergistically elevating the effects of biocontrol and plant promotion through the co-cultivation of the two microorganisms. Our study demonstrated that *T. atroviride* SG3403 is compatible with *B. subtilis* 22. The co-culture metabolites contained a group of antagonistic compounds which were able to inhibit *F. graminearum* growth and increase the activities of pathogen G protein and mitogen-activated protein kinase (MAPK) as compared with axenic culture metabolites. Additionally, the co-culture metabolites enabled us to more significantly decrease the production of gibberellin (GA), deoxynivalenol (DON), and zearalenone (ZEN) from *F. graminearum*, which disorganized the subcellular structure, particularly the cytoplasm of *F. graminearum* hyphae, relative to the axenically cultured metabolites. Furthermore, the seed-coating agent made by the co-culture had significant effects against *F. graminearum* infection by triggering the expression of host plant defensive genes, including *PR1*, *PR3*, *PR4*, *PR5*, *ACS*, and *SOD*. It is suggested that jasmonic acid and ethylene (JA/ET) signaling might dominate wheat’s induced systemic resistance (ISR) against wheat head blight. A dry, powdered bio-seed coating agent containing the co-culture mixtures was confirmed to be a bioavailable formulation that can be applied to control wheat head blight. Taken together, the co-culture’s metabolites or the metabolites and living cells might provide a basis for the further development of a new kind of microbial bio-fungicide in the future.

## 1. Introduction

The microbial community on or inside a plant is a dynamically balanced ecosystem, and the occurrence of plant diseases can break this balance [1]. The introduction of beneficial microbes into pathogen-dominated environments to prevent the emergence of diseases by building a new homeostatic system is generally viewed as the biological control of plant diseases [2]. The application of *Trichoderma* and *Bacillus* to control plant diseases has significantly progressed over the past couple of decades. As one of the world’s three primary biocontrol microorganisms, *Trichoderma* is widely utilized in agriculture, industry, and energy, and has become increasingly important in the biological control of plant diseases—particularly amid governments’ and farmers’ requests for environmentally friendly control measures. It is commonly utilized in the management of soil and airborne diseases [2,3]. *Trichoderma* primarily controls plant diseases by antimicrobial activities, competition, mycoparasitism, and host resistance induction [2]. *Bacillus* microorganisms are Gram-positive bacteria with a high reproduction rate that is widely distributed in air, water, and soil; it has a high tolerance for a variety of environmental stressors, such as UV light, chemical residues, and radiation. *Bacillus amyloliquefaciens*, *B. subtilis*, *B. laterosporus*, and *B. cereus*, for example, are all eco-friendly bacteria that are commonly used as biocontrol agents. They are applied in farming systems to protect against a variety of crop diseases. *Bacillus* has several biological control mechanisms, including competition for nourishment and space with pathogens, the metabolism of antibiotic compounds, the bacteriolysis of pathogens, the induction of plant resistance, and the enhancement of plant growth [4]. *Bacillus*’s antimicrobial proteins, such as iturin, surfactin, and fengycin, can effectively defend against infections, leading to wheat head blight and preventing the development and accumulation of the related toxins [5].

Wheat (*Triticum aestivum* L.), a worldwide food crop, is susceptible to a variety of diseases during its maturation. The prevalence of *Fusarium* head blight in wheat plantations has increased in recent years as a result of global warming [6]. Not only will wheat yield and quality be reduced, but residual toxins will also pose a major threat to human and animal health, which places wheat production in jeopardy. Nivalenol (NIV), deoxynivalenol (DON), and zearalenone (ZEN) are some of the toxins metabolized by *F. graminearum* in wheat grains [7]. Therefore, finding effective control strategies to combat wheat head blight has become a top priority for agricultural products and food safety.

*Trichoderma* and *Bacillus* are well-known. They have some different characteristics that make both available for synergistic applications as biocontrol agents against pathogens; for instance, *Trichoderma* has a relatively stronger mycoparasitism and induction of resistance. *Bacillus* has a stronger antibiotic effect and more tolerance to stress conditions. In this case, to utilize the respective advantages and synergistic effects of both microorganisms, the combined use of these microbes is an optional method to provide a better biocontrol effect. The combination of *B. amyloliquefaciens*, *Paenibacillus polymyxa*, and *B. subtilis* has been demonstrated to lead to the highest inhibition rate against wheat head blight, which could be due to the strains’ synergistic or complementary impact [8]. However, little research has been conducted to improve the biocontrol effects through the co-culturing of *Trichoderma* and *Bacillus*, in which some novel or changed metabolites might be produced through their interactions, having highly effective inhibitory action against multiple pathogen infections.

Co-culturing, which involves the cultivation of two or more microorganisms in the same constrained environment, is partially based on synthetic biology [9]. With this technique, we aimed to enhance a synergistic biocontrol effect. Co-culturing can be accomplished in solid or liquid media, and it is becoming more popular as a method for exploring natural interactions and finding new bioactive metabolites [10]. The differential substances or novel metabolites generated in co-cultures would benefit the synergistic effects between biocontrol microorganisms against plant diseases [11,12]. Therefore, bio-agents based on microbial co-cultures are assured to generate diverse interactions with native microbial communities and plants through signaling or defensive chemicals. This could lead to a further strengthening of the biocontrol.

In this study, we attempted to establish a co-culture system of *T. atroviride* SG3403 and *B. subtilis* 22 through an optimized liquid fermentation approach and to subsequently control the wheat head blight with the co-cultures as seed coating agents.

## 2. Materials and Methods

### 2.1. Fungal Strains

*Trichoderma atroviride* SG3403 (accession number: MN429047) was provided by the Laboratory of Plant Pathology, Shanghai Jiao Tong University. *Bacillus subtilis* 22 (accession number: MN420834) was isolated from soil obtained from the farmland in Qibao town, Shanghai, China. *Fusarium graminearum* (accession number: MN396567) was provided by the Laboratory of Plant Pathology, Shanghai Jiao Tong University. The wheat seeds (cultivar Ningmai 13, moderate resistance levels) were obtained from Nanjing Agriculture University.

### 2.2. Co-Culture and Pathogen Inhibition Assay

*Fusarium graminearum* was grown for 5 days at 28 °C on potato dextrose agar (PDA) plates. The mycelium plugs were taken from the edge of the colony using a punch (diameter 0.5 cm). The plugs were inoculated into a potato dextrose broth (PDB) in a 250 mL flask and then shaken at 28 °C and 180 rpm for 3 days to form abundant *Fusarium* hyphae.

*Trichoderma atroviride* SG3403 was grown on PDA plates for 5 days at 28 °C. The spores were washed with sterile water, and the concentration was adjusted to 2 × 10^8^ spores/mL. *Bacillus subtilis* 22 was cultivated for 2 days at 30 °C on Luria–Bertani (LB) agar plates. The cells were washed in sterile water before being inoculated into LB broth and shaken at 25 °C and 180 rpm until the OD_600_ reached 1.8. Next, 1 mL of *T. atroviride* SG3403 spore suspensions and 1 mL of *B. subtilis* 22 inoculums were inoculated into YMC medium (20 g/mL of yeast powder, 20 g/mL of molasses, 20 g/mL of corn flour, pH of 7.0) at the same time and incubated for 2 d at 28 °C and 180 rpm. Axenic cultures of *T. atroviride* SG3403 and *B. subtilis* 22 inoculums were separately created following the same method as the control treatment. Co-culture or axenic culture metabolites were filtered through a 0.22 μm Millex membrane (Merck, Darmstadt, Hessian, Germany) to remove living cells before use. Then, 1 mL of co-culture or axenic culture metabolites were mixed with 50 mL of PDA (50 °C) at a ratio of 1:50 (*v*/*v*). A *F. graminearum* plug was inoculated at the center of the PDA plate containing co-culture or axenic culture metabolites and cultured at 28 °C for 5 d. The PDA plates without metabolites were used as controls. Each treatment was replicated three times. The diameter of the colony was measured when the control plate was overgrown, and the inhibition rate [13] was calculated as follows.
$$Inhibition\;rate\%=\frac{\left(diameter\;of\;control\;group-diameter\;of\;treatment\;group\right)}{diameter\;of\;control\;group}\times 100$$

*Fusarium graminearum* cultures were inoculated with 1 mL of co-culture or axenic living cells, respectively. The inoculated flasks were incubated for 4–5 days at 28 °C and 180 rpm. Each treatment was replicated three times.

### 2.3. Microscopy Observation

Transmission electron microscopy (TEM) and scanning electron microscopy (SEM) analyses were performed according to the methods previously described with slight changes [14]. *Fusarium graminearum* samples were fixed with 2.5% glutaraldehyde for 6 h at room temperature. Then, the samples were washed with 0.1M PBS buffer (pH 7.0), fixed for 1 h with 1% osmium tetroxide, and dehydrated with an ethanol series. Subsequently, the samples were fixed with an acetone gradient series and embedded in pure epoxy for 8 h. Ultra-thin sections with a thickness of 60–80 nm were cut using an ultramicrotome (Leica EM UC7, Wetzlar, Hessian, Germany). The sections were stained with lead citrate and alcoholic uranyl acetate for 10 min before detecting the copper grids using the TEM (Talos L120C G2, Waltham, MA, USA) and SEM (TESCAN MIRA3, Brno-Kohoutovice, Brno-Kohoutovice, Czech) techniques.

### 2.4. Assays of G Protein and MAPK in Fusarium graminearum

First, 1 mL of *F. graminearum* culture treated with co-culture or axenic culture metabolites was vortexed for 1 min. The solution was centrifuged at 12,000 rpm for 2 min at room temperature; the supernatant was taken, and the microbial G protein (GTP) and microbial mitogen-activated protein kinase (MAPK) were analyzed with the appropriate ELISA detection kits (Biosamite, Shanghai, China), according to the manufacturer’s instructions. Each treatment was replicated three times.

### 2.5. Assays of GA and Toxins Production in Fusarium graminearum

First, 1 mL of *F. graminearum* culture treated with co-culture or axenic culture metabolites was vortexed for 1 min. The solution was centrifuged at 12,000 rpm for 2 min at room temperature, and the supernatant was taken for toxin detection. A gibberellin (GA) ELISA detection kit (Biosamite, Shanghai, China) was applied to determine the content of GA in the pathogen according to the manufacturer’s instructions. The UHPLC-MS/MS was used for the detection of the toxins of DON and ZEN. The cultures were collected using centrifugation at 8000 rpm, and the mycelia were extracted with methanol. Then, the supernatant and extract were mixed and concentrated under reduced pressure using a rotary evaporator at 65 °C. Subsequently, methanol/water (80/20, *v*/*v*) was used for dissolution, filtration through a 0.22 µm nylon filter was performed, and the solution was stored in brown glass vials at −20 °C until further detection. The toxin sample was detected using a Sciex Quadrupole 5500 instrument (Waters, Milford, MA, USA) on a C8 column. The mobile phase was a 1% ammonium formate solution. The scanning modes were positive and negative ion scanning modes, and the detection mode was multi-reaction monitoring. Data acquisition and processing were performed using MassLynx v4.1 and Quanlynx software (Waters, Milford, MA, USA). Each treatment was replicated three times.

### 2.6. Seed Treatments with Biocontrol Microbial Culture

To prepare a liquid bio-seed coating agent (SCA), the co-culture and axenic culture containing living cells of *T. atroviride* and *B. subtilis* and the film-forming agent (Jilin Bada Pesticide Co. Ltd., Shanghai, China) were mixed at a ratio of 8:1:1 (*v*/*v*), respectively, and a film-forming agent treatment without any microbe culture was used as the control. The dry powder bio-SCA was prepared by using Mukherjee’s method [15], modified as described. The co-culture or axenic culture and talc powder were mixed at a ratio of 1:5 (mL:g) and dried at 40 °C for 6 h; the talc powder treatment without any microbes was used as the control. Wheat seeds were uniformly coated with the liquid SCA and dry powder SCA following the ratios of agent to seeds of 1:50 (mL:g) and 1:100, respectively. After being left at room temperature for 2 h, the seeds were placed in a dark incubator at 25 °C, and the germination rate was measured after 36 h. Each treatment was repeated three times.

Coated wheat seeds were sown at 10 seeds per pot, and pots were placed in a greenhouse with a light and dark cycle of 14 h/10 h, at 25 °C, for 14–21 days. Three seedlings were selected at random from each pot in all treatments, and the plant heights and fresh weights were measured.

For the analysis of the biocontrol efficiency of the bio-SCA against *Fusarium* head blight, wheat ears were treated by spraying 1 L of 5 × 10^7^ CFU/mL of biocontrol agent suspension per treatment when wheat plants came to the flowering stage. The symptom of *Fusarium* head blight was observed 20 days after pathogen inoculation. The experiment was repeated three times. Relative disease severity and biocontrol efficacy were calculated with the following formula. Wheat was scored by the disease index [16] (DI) on a scale of 0–9, as follows: no symptoms (DI = 0); 1–10% of the wheat ears infected (DI = 1); 11–25% of the wheat ears infected (DI = 3); 26–50% of the wheat ears infected (DI = 5); 51–75% of the wheat ears infected (DI = 7); 76–100% of the wheat ears infected (DI = 9).
$$Relative\;disease\;severity\%=[\sum \frac{\left(The\;number\;of\;diseased\;plants\;in\;each\;grade\times number\;of\;plants\right)}{\left(total\;number\;of\;plants\times highest\;disease\;index\right)}\times 100$$
$$Biocontrol\;efficacy\%=\left[\frac{\left(Relative\;disease\;severity\;of\;control\;treatment-disease\;severity\;of\;treatment\right)}{disease\;severity\;of\;control}\right]\times 100$$

### 2.7. Gene Expression Study

*Fusarium graminearum* cultures were collected after 4–5 days of inoculation. The grains and leaves of each treatment were collected after *F. graminearum* was inoculated for 14 to 21 days. The extraction and purification of total RNA from plant tissues was performed with a FastPure Plant Total RNA Isolation Kit (Vazyme, Shanghai, China). Synthesis of single-stranded cDNA RT-qPCR used the HiScript III 1st Strand cDNA Synthesis Kit (Vazyme, China). The RT-qPCR reaction was carried out with a ChamQ Universal SYBR qPCR Master Mix kit (Vazyme, China) in a Roche lightcycle95 (Basel, Switzerland). The target genes of the wheat to be detected corresponded to: the disease-related proteins PR1, PR3, PR4, and PR5; the ethylene synthesis-related gene *ACS*; and the antioxidant enzyme gene *SOD*. The reference gene was *β-actin*. The specific primers are shown in Table 1, which was completed by Qingke Biotechnology (Beijing, China). Each treatment was replicated three times.

### 2.8. Field Assay

A field experiment was conducted at Shanghai Jiao Tong University (SJTU). Three parallel experimental groups involving the dry powder seed-coating agent (SCA) of *T. atroviride* and *B. subtilis*, liquid SCA of *T. atroviride* and *B. subtilis*, and control groups used a 90 m^2^ wheat field.

A field experiment was also conducted at the Zhumadian Academy of Agricultural Sciences (ZAAS). The field was mainly used for screening wheat biocontrol agents. The crops had been continuously planted for many years, and the wheat head blight was very severe in this field. Three parallel experimental groups, named the TB (*T. atroviride* and *B. subtilis* dry powder SCA), CA (50% carbendazim), and control groups, took up a 180 m^2^ wheat field. Before planting, the wheat seeds were treated with 150 g/mu (1 mu = 666.67 m^2^ approximately) of seed-coating agents (5 × 10^8^ CFU/g) or 30 g/mu of carbendazim, respectively.

Another field experiment was conducted at Xinxiang, Henan province, China. Three parallel experimental groups, TB (*T. atroviride* and *B. subtilis* dry powder SCA), CA (Tebuconazole), and control, occupied a 180 m^2^ wheat field. Before planting, the wheat seeds were treated with 300 g/mu of wettable powder (5 × 10^8^ CFU/g) and 10 g/mu of tebuconazole. The field assay was repeated three times. Five points were sampled in each plot, and 200 ears were randomly surveyed at each point. Relative disease severity and biocontrol efficacy were calculated. The wheat yield increase rate was calculated with the following formula.
$$Wheat\;yield\;increase\%=\left[\frac{\left(wheat\;yield\;of\;treament-wheat\;yield\;of\;control\right)}{wheat\;yield\;of\;control}\right]\times 100$$

## 3. Results

### 3.1. Inhibition of Fusarium graminearum Growth by Co-Culture Metabolites

The SEM results (Figure 1) showed that *B. subtilis* adhered to the surface of *T. atroviride* in co-cultures, indicating that the two strains had a compatible relationship with each other. To clarify the inhibitory effect of the co-culture metabolites of *B. subtilis* and *T. atroviride* on *F. graminearum*, the filtrate was mixed with PDA and tested, and the antimicrobial effect of the co-culture filtrate reached 61.5%, which was significantly (*p* < 0.001) higher than that of the axenic culture filtrate. The inhibitory rates of *B. subtilis* and *T. atroviride* were 42.9% and 5.2%, respectively. Hence, the co-culture metabolites of *B. subtilis* and *T. atroviride* exhibited a synergistic effect on the inhibition of *F. graminearum*.

### 3.2. Effects of Co-Culture Metabolites on the Mycelium Structure of Fusarium graminearum

The co-cultured metabolites caused the hyphae of *F. graminearum* to have thinner and malformed cell walls, as compared with the axenic culture treatment and untreated hyphae (Ck). Thus, the co-culture resulted in a greater degradation of the pathogen’s cell wall compared with the axenic culture (Figure 2A). Furthermore, transmission electron microscopy (TEM) revealed that the hyphal cytoplasm became wholly degraded and disorganized relative to the axenic culture treatment and CK. Due to the co-culture treatment, the hyphal cell membranes of *F. graminearum* became significantly plasmolyzed, the cytoplasm was massively degraded, and the big vacuole-like structures were also diminished. However, the outer layer of the cell wall was destroyed or more severely eroded in the axenic culture treatment of *B. subtilis* and *T. atroviride* (Figure 2B).

### 3.3. Effects of Co-Culture Metabolites on the Activity of MAPK and G Protein of F. graminearum

MAPK and G protein are crucial signaling molecules that are intimately involved in *F. graminearum*’s growth and development. Therefore, we needed to understand what impacts were generated by a co-culture of *Trichoderma* and *Bacillus* on the activities of both signaling molecules. A quantitative analysis of the MAPK activity in each treatment revealed that the co-culture metabolite treatment induced an increase of more than 20% compared with the axenic culture (Figure 3A). It was found that the co-culture metabolites also caused an abnormal change in G protein activity. *F. graminearum* had higher G protein activity after treatment with the co-culture as compared to treatments with the *B. subtilis* culture and *T. atroviride*, but still less than that of the control treatment. Thus, MAPK and G protein appear to be incited by the co-culture metabolites during the microbial interaction process. We inferred that the increased activities are a type of the pathogen’s response to the co-culture’s action.

### 3.4. Effects of Co-Cultured Living Cells on the Production of GA and Mycotoxins by F. graminearum

The ability to synthesize GA phytohormones was tested in the liquid shaking cultures of *F. graminearum* mixed with the co-culture’s and axenic culture’s living cells. The concentration of GA in the culture fluids of the co-culture’s living cells was slightly decreased as compared with the control (Figure 4A). However, the axenic culture’s living cells had no inhibitory effect on the production of GA, and the yield was even higher than that of the control.

To investigate the effects of the axenic culture’s and co-culture’s living cells on *F. graminearum* mycotoxin production, the amounts of ZEN and DON produced were examined. The DON content after the treatment with the co-culture’s living cells saw a reduction of about 76.6% (Figure 4A,B). The *B. subtilis* culture had higher activity than *T. atroviride* in the reduction of *F. graminearum* DON production, but the synergistic action of both organisms was clearly shown in the reduction of the pathogen’s DON production. Our experiments proved that the inhibition rate of ZEN production ranged from 78.4% to 90.3% (Figure 4A,B) due to the treatment with the co-culture’s living cells or the axenic culture’s living cells. The co-cultivation treatment had a more effective inhibitory effect on the generation of the ZEN toxin of *F. graminearum*. In addition, the expression trends (Figure 4D) of DON (Tri5) and ZEN (ZEA1 and ZEA2) biosynthesis genes were similar to the UHPLC results. This indicates that the co-culture’s living cells had a synergistic inhibitory effect on the synthesis of DON and ZEN in *F. graminearum.*

### 3.5. Effects of Bio-Seed Coating Agent on Wheat Seed Germination and Plant Growth

Different formulations of bio-seed coating agents were prepared: *B. subtilis*, *T. atroviride*, co-culture, and control (without both microorganisms). In the treatment of liquid bio-seed coating agents, there was a slight inhibitory impact of bio-seed coating agent treatment on seed germination, particularly in the axenic culture treatment; however, there was no significant difference among them (Figure 5A). Comparatively, dry powder bio-seed coating generated a better improvement in seed germination than liquid bio-seed coating agent, and the difference was not significant (Figure 5A,D). In general, bio-seed coating agents from either the axenic culture or the co-culture had no significant negative influence on wheat seed germination; the dry powder seed-coating agent was able to improve wheat seed germination.

Coating seeds with the co-culture as a liquid seed-coating agent was better for seedling growth than using the axenic culture as the liquid seed-coating agent (Figure 5B). The liquid bio-seed coating agent prepared with either *T. atroviride* or *B. subtilis* was better for seedling growth than the dry powder with either microorganism (Figure 5B). However, the dry powder prepared from the co-culture was better for seedlings’ growth than the others (Figure 5B,E). Similarly, the liquid formulation of the co-culture exerted a slightly better effect on seedling growth (Figure 5C). Thus, the dry powder seed-coating agent prepared with the co-culture’s living cells has been demonstrated to be an efficient bio-seed coating agent.

### 3.6. Effects of Bio-Seed Coating Agent against Wheat Head Blight

The infection inhibition effects of the dry powder seed-coating agents prepared from *B. subtilis*, *T. atroviride*, and the co-culture against *F. graminearum* were 28.4%, 13.4%, and 66.2% (Figure 5F), respectively. Additionally, the effects of the liquid seed-coating agent were 10.9%, 20.4%, and 45.2% (Figure 5F), respectively. The bio-seed-coating agent prepared from the co-culture’s living cells provided better control of the wheat head blight as compared with the axenic culture of either *B. subtilis* or *T. atroviride*. In comparison, the dry powder seed-coating agents were more available for disease control.

### 3.7. Effects of Seed Treatment on the Expression of Defense Genes in Wheat

The expression levels of the pathogenesis-related proteins (*PR1*, *PR3*, *PR4*, and *PR5*), ethylene synthesis pathway (ACS), and superoxide dismutase (SOD)-related genes in greenhouse wheat plants are shown in Figure 6. For *PR1*, there was no significant difference between the axenic culture and co-culture treatments (Figure 6A); even though the *T. atroviride* culture showed higher stimulation of *PR1*, the *PR3* gene expression was only slightly higher in the co-culture treatment group (Figure 6B). The *PR4*, *PR5*, *ACS*, and *SOD* genes in plants were upregulated only in co-culture mixtures, indicating a significant synergistic action of the two microorganisms (Figure 6C,E,F). Overall, the bio-seed coating agents prepared with either the axenic culture or a co-culture of both microorganisms were able to activate wheat defense gene expression; however, the synergistic action of the co-culture was most significantly shown in the activation of the expression of *PR4*, *PR5*, and *SOD* in wheat.

### 3.8. Effects of Seed-Coating Agents on Yield and Control of Wheat Head Blight in Field Assays

Two types of seed-coating agents, *T. atroviride* SG3403 and *B. subtilis* 22, were shown to increase the control of wheat head blight, and the dry powder formulation of the combined agent was better than the liquid formulation (Figure 7A,B). Then, the dry powder seed-coating agent of *T. atroviride* SG3403 and *B. subtilis* 22 co-culture mixtures (TB) and the 50% carbendazim (CA) agent were selected for the field assay to further verify the above results. In terms of control effect, TB2 showed the highest, with an average of 46.33%, and chemical agents followed with an average of 30.89% (Figure 7C). According to statistical analysis, there was a significant difference between TB and CA. The yield of each treatment was higher than the control yield. The yield increase thanks to TB was 13.59% compared with the chemical agent treatment (Figure 7D).

The dry powder seed-coating agent of the *T. atroviride* SG3403 and *B. subtilis* 22 co-culture mixture (TB) and tebuconazole (CA) were selected for the field assay to further verify the above results (Figure 8A). In terms of disease control, CA was the stronger than TB, with an average of 37.02%, and TB followed with an average of 27.32% (Figure 8B). According to statistical analysis, there was no significant difference between the TB and CA treatments. The yield of each treatment was higher than that of the control. The yield increases from TB and CA were 5.08% and 6.81%, respectively, compared with the control (Figure 8C). Although the control effect of TB was only 73.80% that of CA, the use of chemical agents could be reduced by using the TB seed-coating agent. It was demonstrated that the co-culture seed-coating agent of *Trichoderma* and *Bacillus* had a synergistic effect on the control of wheat head blight and can also improve wheat production. Furthermore, the usage of *T. atroviride* SG3403 and *B. subtilis* 22 can eliminate environmental pollution caused by chemical agents.

## 4. Discussion

Microbial co-cultivation technology is widely used for changing microbial metabolism, thereby generating a range of novel metabolites via inter-microbial competition and communication [17]. Recently, co-cultivation has represented great interest as a way to enact metabolic system alterations. The use of microbial consortiums in agriculture may increase microbe effectiveness, stability, and uniformity in plant disease prevention and plant growth promotion in a variety of soil and environmental circumstances [18]. As a consortium of biological control agents (BCAs), *Trichoderma asperellum* and *B. amyloliquefaciens* have several positive effects, including pathogen protection, growth promotion, disease resistance induction, and generation of bioactive substances with antibiotic and anticancer characteristics [19,20]. However, the majority of the consortium of *Trichoderma* and *Bacillus* are simply prepared by mixing separative cultures of each strain instead of co-cultures prepared in the same fermentation tank. The previous investigation showed that the co-culturing of *B. amyloliquefaciens* and *T. asperellum* was able to significantly inhibit the pathogen *B. cinerea*’s growth [21]: a range of specific or upregulated antibiotic substances generated in co-cultures are supposed to be crucial in pathogen inhibition. A co-culture of *Trichoderma* and *Bacillus* provides an alternative for the development of new kinds of microbial bio-pesticides or bio-manures. In this study with *T. atroviride* SG3403 and *B. subtilis* 22, it was important to demonstrate whether the co-culturing of these microorganisms generated synergistic effects on pathogen inhibition, pathogen toxin degradation, wheat seed germination, seedling growth, and head blight control.

The co-culture of *T. atroviride* SG3403 and *B. subtilis* 22 was confirmed to have strong synergistically improved action in wheat growth promotion, pathogen inhibition, and disease control. Interestingly, *F. graminearum*’s cell wall, cytoplasm, and subcellular organelles were more severely damaged by the co-culture than by the axenic culture, indicating that some potential cell wall degrading enzyme (CWDE) or antibiotic substances are specifically produced due to the interaction of both microorganisms in a co-culture [5]. For instance, the antimicrobial effect of the co-cultured fermentation liquor of *B. amyloliquefaciens* ACCC11060 and *T. asperellum* GDFS1009 was found to be significantly higher than that of pure cultivation of a single strain. An increase in the synthesis of antimicrobial substances contributed to this result [21]. Co-cultivation of *Trichoderma* and *Bacillus* produces more abundant secondary metabolites, including those that promote plant growth, such as indole-3-butyric acid and 1-naphthyl acetic acid [22].

When serious pathogen infection occurs, overproduction of GA from *F. graminearum* will lead to abnormal seed germination and host plant growth, including in wheat and rice [23]. The GA content of the co-culture treatment was significantly lower than those of the other treatments in this study. This finding might be valuable for effectively limiting GA’s damage to crop growth. Mycotoxins are secondary metabolites produced from pathogenic fungi, which generated significant toxic effects on plants, animals, and humans [24]. DON (deoxynivalenol) and ZEN (zearalenone) are major toxins from *F. graminearum* causing wheat head blight. They contaminate wheat grains and feedstuffs. The biodegradation process is regarded as the ideal alternative approach for toxin detoxification [25]. The effects of co-cultures of *T. atroviride* and *B. subtilis* were more effectively shown in the reduction of DON and ZEN production. It was suggested that the formation of toxin-degrading enzymes peroxidase, zearalenone phosphotransferase, and UDP-glucuronosyltransferase was enhanced in the co-culture [25,26,27,28,29,30].

The G protein is known as a transmembrane signal transducer that is important in fungi development [31]. In the study, the co-culture stimulated greater G protein activity in *F. gramiearum* than the other treatments. Because G protein is demonstrated to promote sporulation of *Trichoderma* and toxin degradation, we inferred that the G protein increase in *Fusarium* mycelia might be a regulatory mechanism potentially utilized by *Trichoderma* to adhere to the pathogen for sporulation. In addition, a previous study demonstrated that the two G subunits of GzGPA1 and GzGPB1 cause negative regulation in the synthesis of DON and ZEN in *F. graminearum*. This is probably one of the reasons why both toxins declined in the co-culture treatment [32,33]. The MAPK activity of *F. graminearum* in the co-culture treatment was also found to be higher than in the other treatments. Additionally, improved MAPK activity in *F. graminearum* would also benefit *Trichoderma* sporulation and mycoparasitism, particularly through its interaction with the pathogen, resulting in a higher effectiveness of *Trichoderma* against the pathogen’s growth and reproduction.

The co-cultivation metabolites of *T. asperellum* GDFS1009 and *B. amyloliquefaciens* 1841 have been demonstrated to improve wheat growth, including root length and plant height [22]. Coating seeds with *B. subtilis* and *T. asperellum* offers protection to the seedlings against damping-off disease and results in improved growth [34]. In this study, two formulations of the best bio-seed coating agent, in dry powder and liquid forms prepared with co-culture, promoted wheat growth in the pot and filed assays. Dry powder seed coatings outperformed liquid seed coating in terms of germination and growth stimulation in the experiments. Talc in dry powder seed-coating agents, as inert and stable substances, allow the seed-coating agent more tolerance to a stressful environment. Meanwhile, the film-forming agent in the liquid seed-coating agent has complex chemical properties, which probably have an impact on seed germination and plant growth in complicated ways [35]. Beyond the mentioned reasons, from the perspective of application, the dry powder bio-seed coating agent is easy to package, store, transport, and use in a farming system [35]. In addition, the co-culture wettable powder and seed-coating agent of *T. atroviride* SG3403 and *B. subtilis *22 have the same effects on wheat head blight control and growth promotion.

The salicylic acid (SA, jasmonic acid (JA), and ethylene (ET) pathways are all involved in disease resistance signaling, and the SA and JA/ET defensive pathways are thought to be mutually antagonistic [36,37,38]. In our work, the SA pathway marker genes *PR1* and *PR5*, the JA pathway marker genes *PR3* and *PR4*, and the rate-limiting enzyme ACS from the ethylene synthesis pathway were used for evaluating the ability of the co-culture to systemically induce resistance against wheat head blight [39]. We demonstrated that the defense signaling effect could be transferred from the wheat seed and root up to the wheat ear and leaf at the manure stage of development. Generally, *PR3*, *PR4*, and *ACS* in the wheat plant were highly and synergistically activated by the co-culture, indicating that the JA/ET pathway is more inducible by the co-culture instead of the axenic culture of either microorganism. *SOD* gene expression in the wheat plant was specifically upregulated by seed treatment with the co-culture agent, which means the co-culture can increase wheat’s resistance to a harsh environment and also to oxidative stress. *SOD* can help plants remove peroxidative free radicals, leading to protection of the cell membrane system, particularly during pathogen infection [40,41]. Therefore, the co-culture seed-coating agent is able to systematically induce wheat plants’ resistance against *Fusarium* head blight through the activation of salicylic acid (SA), jasmonic acid (JA), ethylene (ET), and reactive oxygen species (ROS).

The filed trial of the seed-coating agent followed. To the best of our knowledge, this is the first report on a seed-coating agent using a co-cultivation of *T. atroviride* and *B. subtilis* against *Fusarium* head blight in the field. The co-culture seed-coating agent of *Trichoderma* and *Bacillus* had a synergistic effect on the control of wheat head blight and can also improve wheat production. So far, much research on microbial seed coating has been conducted with vegetables, cereals, oilseeds, and legumes. Most of these studies found that after application of a PGPM (plant growth promoting microorganism) to crops via seed coating, the survival of seeds and effects on seed germination, seedling establishment, nodulation (in legumes), plant growth, yield, plant disease protection, and even nutritional value were usually positive [42]. Furthermore, the usage of *T. atroviride* SG3403 and *B. subtilis* 22 can eliminate environmental pollution caused by chemical agents.

## 5. Conclusions

The co-cultivation of *T. atroviride* SG3403 and *B. subtilis* 22 can suppress *F. graminearum* growth and inhibit the synthesis of DON and ZEN toxins. The improved expression of G protein and MAPK in *F. graminearum* by co-culture might be one of the mechanisms that can make *T. atroviride* SG3403 and *B. subtilis* 22 more effective against *F. graminearum*. Additionally, the co-culture agent can also maintain the normal growth of the wheat by minimizing the excessive accumulation of gibberellins (GA) if the pathogen seriously infects the host. Co-culture mixtures based on dry powder seed-coating agents can improve wheat seed germination, wheat growth, and defense against *F. graminearum* infection. Taken together, co-cultivation of *T. atroviride* SG3403 and *B. subtilis* 22 provides a novel way to innovate microbial pesticides or manure.

## Figures and Tables

**Figure 1 jof-08-01250-f001:**
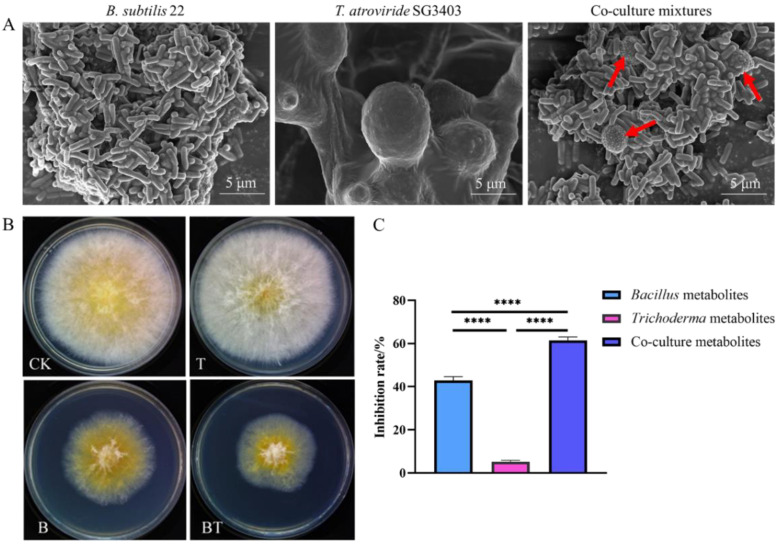
Compatible interaction of *B. subtilis* and *T. atroviride* and effect of the co-culture filtrate of *B. subtilis* and *T. atroviride* on *F. gramiearum* growth. (**A**) SEM results with respect to the co-culture interaction; the red arrow denotes the mycelia of *T. atroviride*. (**B**) Inhibitory effects of the co-culture and axenic culture metabolites on *F. graminearum*. B denotes axenic culture metabolites of *B. subtilis*; T denotes axenic culture metabolites of *T. atroviride*; BT denotes co-culture metabolites. (**C**) Inhibitory rates of the co-culture and axenic culture metabolites on *F. graminearum*. The significance of the difference between the axenic and co-culture metabolites was *p =* 0.001. **** *p* ≤ 0.001.

**Figure 2 jof-08-01250-f002:**
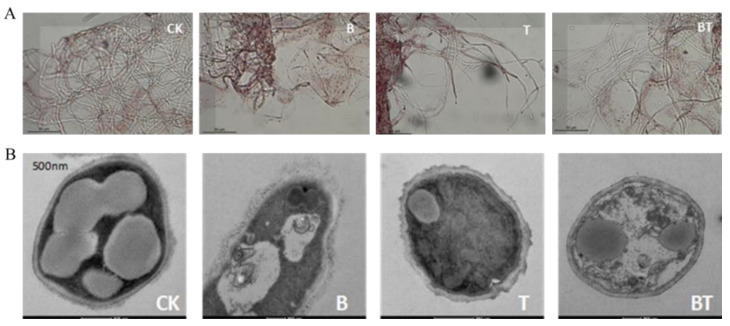
Effects of co-culture and axenic culture metabolites on (**A**) the structures of mycelia and (**B**) mycelia ultra-structures of *F. graminearum*. CK denotes control; B denotes axenic culture metabolites of *B. subtilis*; T denotes axenic culture metabolites of *T. atroviride*; BT denotes co-culture metabolites.

**Figure 3 jof-08-01250-f003:**
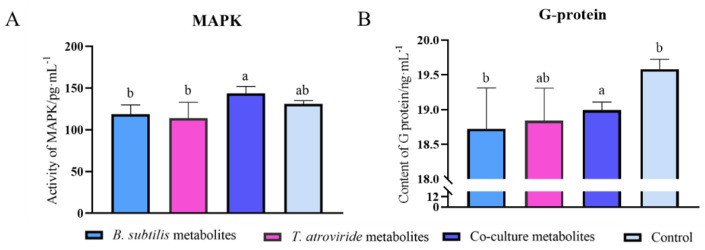
Impact of co-culture filtrate on the activity of (**A**) the mitogen-activated protein kinase and (**B**) G protein receptor 1 (GPR1). Data resulted from biological triplicate cultures with microplate reader technical duplicates. a, b, represent significant differences between the axenic and co-culture treatments (*p* < 0.05).

**Figure 4 jof-08-01250-f004:**
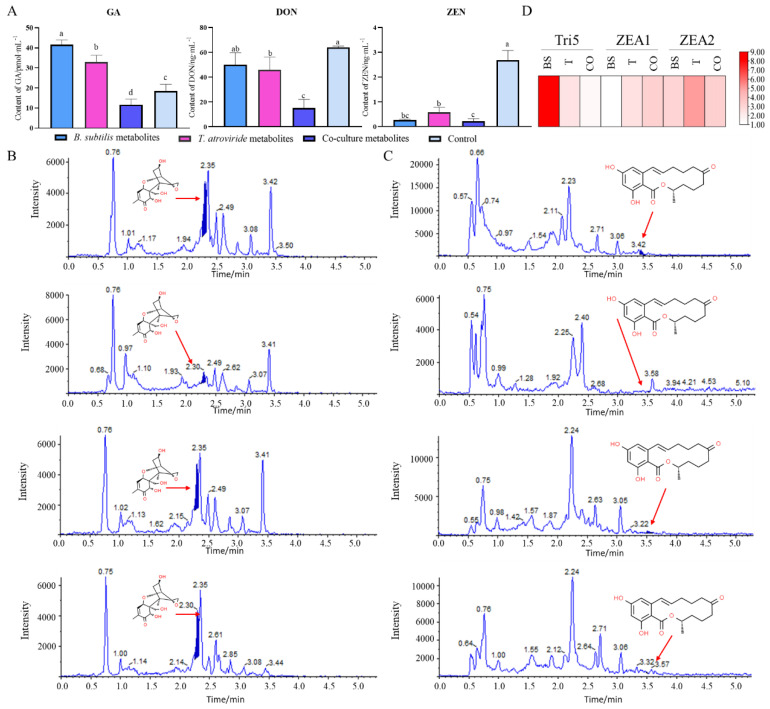
Effects of the co-culture and the axenic culture of *T. atroviride* and *B. subtilis* on the production of (**A**) gibberellin (GA), deoxynivalenol (DON), and zearalenone (ZEN) in *F. graminearum*. UHPLC-MS/MS analysis results of (**B**) DON and (**C**) ZEN. The co-culture’s living cells had a synergistic inhibitory effect on synthesis of GA, DON, and ZEN in *F. graminearum*. (**D**) Relative expression levels of genes related to DON (Tri5) and ZEN (ZEA1 and ZEA2) biosynthesis genes, each normalized to GAPDH. BS denotes *B. subtilis* treatment; T denotes *T. atroviride* treatment; CO denotes co-culture treatment. a, b, c, d represent significant differences between the axenic and co-culture treatments (*p* < 0.05).

**Figure 5 jof-08-01250-f005:**
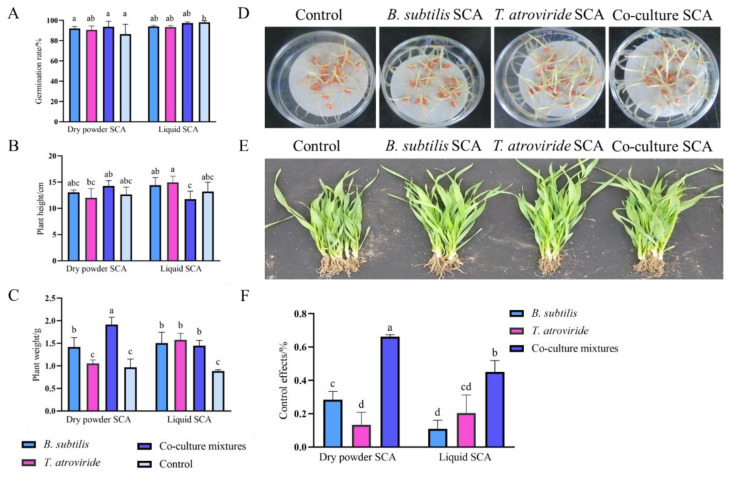
Effects of co-culture and axenic culture seed-coating agent (SCA) on: (**A**) the germination rate of wheat seeds; (**B**) plant height of wheat; (**C**) raw weight of wheat seedlings; (**D**) dry powder seed-coating agent treated wheat seeds; (**E**) dry powder seed-coating agent-treated wheat seedling. (**F**) Control effects of *T. atroviride*, *B. subtilis*, and co-culture mixtures against *F. graminearum*. Wheat seeds were treated with dry powder and a liquid seed-coating agent in this study. The dry powder seed-coating agent has been demonstrated to be an efficient SCA. a, b, c, d represent significant differences between the axenic and co-culture treatments (*p* < 0.05).

**Figure 6 jof-08-01250-f006:**
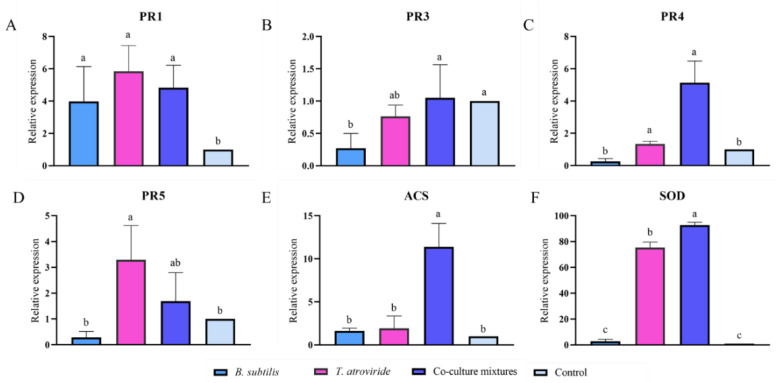
Expression of pathogenesis-related genes (**A**) *PR1*, (**B**) *PR3*, (**C**) *PR4*, and (**D**) *PR5*; and (**E**) ethylene synthesis pathway (*ACS*) and (**F**) superoxide dismutase (*SOD*)-related genes in wheat after the application of co-culture and axenic culture seed-coating agents. The synergistic action of co-culture was shown most significantly in the activation of the expression of the *PR4*, *PR5*, and *SOD* genes in wheat. a, b, c represent significant differences between the axenic and co-culture treatments (*p* < 0.05).

**Figure 7 jof-08-01250-f007:**
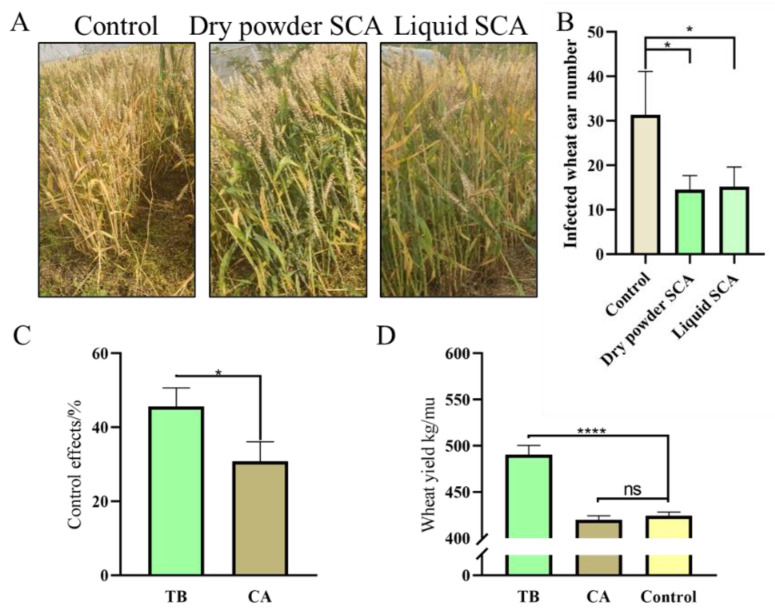
Effects of different types of seed-coating agent (SCA) on (**A**,**B**) number of wheat head blight-infected wheat ears in SJTU crop. The two types of SCA were prepared with the co-culture mixtures of *T. atroviride* SG3403 and *B. subtilis *22. Effects of *Trichoderma*–*Bacillus* SCA on (**C**) control of wheat head blight and (**D**) wheat production in ZAAS. TB denotes the seed-coating agent of *T. atroviride* SG3403 and *B. subtilis* 22 co-culture mixtures and CA denotes the 50% carbendazim. The co-culture seed-coating agent of *Trichoderma* and *Bacillus* showed a synergistic effect on the control of wheat head blight and could also improve wheat production. *T. atroviride* SG3403 and *B. subtilis* 22 had a better effect together. * represents significant differences between the treatment and control (*p* < 0.05). **** represents significant differences between the treatment and control (*p* < 0.001). “ns” represents non-significant differences between the treatment and control.

**Figure 8 jof-08-01250-f008:**
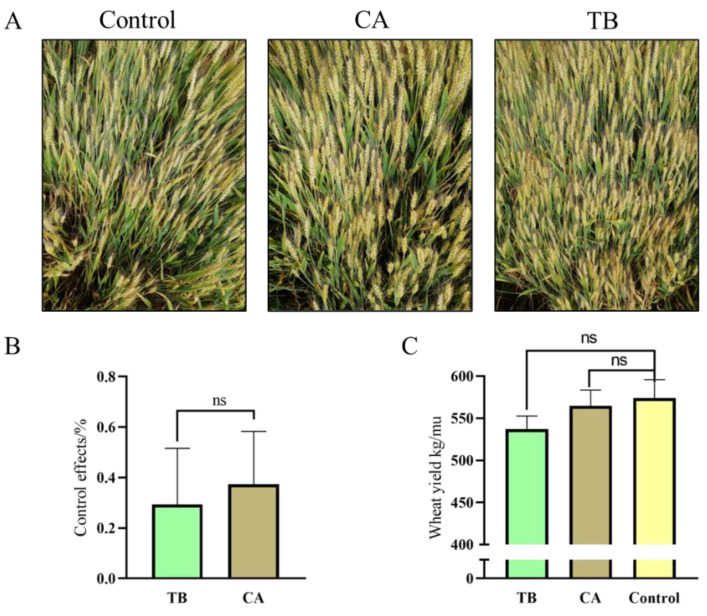
Effects of *Trichoderma*–*Bacillus* SCA on (**A**,**B**) control of wheat head blight and (**A**,**C**) wheat production in Xinxiang, Henan Province, China. TB denotes the seed-coating agent of *T. atroviride* SG3403 and *B. subtilis* 22 co-culture, and CA denotes the tebuconazole. The co-culture seed-coating agent of *Trichoderma* and *Bacillus* has a synergistic effect on the control of wheat head blight and can also improve wheat production. The disease control effects of *T. atroviride* SG3403 and *B. subtilis* 22 were similar to that of the chemical agent. Additionally, the impacts on production were the same. This demonstrated that the TB agent can reduce the use of chemical fungicides. “ns” represents non-significant differences between the treatment and control.

**Table 1 jof-08-01250-t001:** Primers for RT-qPCR.

Name	Forward Primer (5′-3′)	Description
*PR1*	F-CTACGACTACGGGTCCAACA R-GCTTATTACGGCATTCCTTT	Salicylic acid Pathway
*PR3*	F-AGAGATAAGCAAGGCCACGTC R-GGTTGCTCACCAGGTCCTTC	Jasmonic acid Pathway
*PR4*	F-CGAGGATCGTGGACCAGTG R-GTCGACGAACTGGTAGTTGACG	Jasmonic acid Pathway
*PR5*	F-ACAGCTACGCCAAGGACGAC R-CGCGTCCTAATCTAAGGGCAG	Salicylic acid Pathway
*SOD*	F-GCCTTTTGGCCTCTTTATCC R-AACCTCAAGCCCATCAGCG	Antioxidant enzyme gene
*ACS*	F-CTCTCGCTGGACCTGATCG R-GTCCTGGAAATTGGCGATCC	Ethylene Pathway
*β-actin*	F-CTCTGACAATTTCCCGCTCA R-ACACGCTTCCTCATGCTATCC	Housekeeping gene of wheat
*Tri5*	F-TCTATGGCCCAAGGACCTGTTTGA R-TGACCCAAACCATCCAGTTCTCCA	DON biosynthesis-related gene
*ZEA1*	F-GGCACTTTGACAACCGCTTC R-TTGCGCCGTCTGAGTACCC	ZEN biosynthesis-related gene
*ZEA2*	F-ACATGCGTGTCGTTGTTGTAAG R-TGCACCGTGAGAAGCCGT	ZEN biosynthesis-related gene
*GAPDH*	F-CTACATGCTCAAGTACGACTCTTCC R-GCCGGTCTCGGACCACTTG	Housekeeping gene of *Fusarium graminearum*

Note: F denotes forward primer; R denotes reverse primer.

## Data Availability

Not applicable.

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
