# Peer review of "Effects of Trichoderma atroviride SG3403 and Bacillus subtilis 22 on the Biocontrol of Wheat Head Blight"

_jof, 2022, doi:10.3390/jof8121250_

Round 1
Reviewer 1 Report (Previous Reviewer 1)
Dear authors,
Thank you for your reply to my comments.
Author Response
Dear Ms. Norah Liu and dear reviewers
Re: Title: "Effect of Trichoderma atroviride SG3403 and Bacillus subtilis 22 in the biocontrol of wheat head blight " (original ID: jof-1905553)
Thank you for your letter and the reviewers’ comments concerning our manuscript entitled “Effect of Trichoderma atroviride SG3403 and Bacillus subtilis 22 in the biocontrol of wheat head blight”. Those comments are valuable and very helpful. We have read through comments carefully and have made corrections. And english editing was finished by MDPI. Based on the instructions provided in your letter, we uploaded the file of the revised manuscript. Revisions in the text are shown using red highlight for additions, and strike through font for deletions.
We would love to thank you for allowing us to resubmit a revised copy of the manuscript and we highly appreciate your time and consideration.
Sincerely.
Hongyi Liu
Corresponding author:
Prof. Dr. Jie Chen
School of Agriculture and Biology, Shanghai Jiao Tong University
Shanghai 200240, China
E-mail: jiechen59@sjtu.edu.cn
Reviewer 2 Report (New Reviewer)
The manuscript entitled "Effect of Trichoderma atroviride SG3403 and Bacillus subtilis 22 in the biocontrol of wheat head blight" presents timely and adequate information that will be relevant to the readers. However, I feel that the manuscript needs some improvement regarding the writting:
- There are some sentences that are too long and difficult to understand;
- Other sentences are too confusing and make no sense;
- Sometimes, some sentences are repeated twice;
- Some species names are not in italics;
- From my point of view, the introduction and discussion do not include relevant references, as there are sentences that lack a scientifically support.
I also recomend the authors to verify thal all gene names are in italics, so the manuscript is troughly concise. The reference list must also be checked carefully.
All my corrections and suggestions are in the attached pdf.
Kind Regards.

Author Response
Dear Ms. Norah Liu and dear reviewers
Re: Title: "Effect of Trichoderma atroviride SG3403 and Bacillus subtilis 22 in the biocontrol of wheat head blight " (original ID: jof-1905553)
Thank you for your letter and the reviewers’ comments concerning our manuscript entitled “Effect of Trichoderma atroviride SG3403 and Bacillus subtilis 22 in the biocontrol of wheat head blight”. Those comments are valuable and very helpful. We have read through comments carefully and have made corrections. And english editing was finished by MDPI. Based on the instructions provided in your letter, we uploaded the file of the revised manuscript. Revisions in the text are shown using red highlight for additions, and strike through font for deletions.
We would love to thank you for allowing us to resubmit a revised copy of the manuscript and we highly appreciate your time and consideration.
Sincerely.
Hongyi Liu
Corresponding author:
Prof. Dr. Jie Chen
School of Agriculture and Biology, Shanghai Jiao Tong University
Shanghai 200240, China
E-mail: jiechen59@sjtu.edu.cn

Round 2
Reviewer 2 Report (New Reviewer)
Dear authors,
Thank you for your effort to improve the manuscript. I really appreciate that you have accepted my suggestions. Congrats on this manuscript!
I have some minor comments:
Lines 98, 100, 104, 105: Please add the medium brand, city and country. Just an example: PDA (Merck, Dramstadt, Germany).
Line 262: Fusarium graminearum
Please check all references. Some species names are wrong. It is Fusarium graminearum and not Fusarium Graminearum. And some Journals are not abbreviated nor in italics. I recommend the authors to check this, otherwise, the Journal team will ask you to do the same.
Kind Regards
This manuscript is a resubmission of an earlier submission. The following is a list of the peer review reports and author responses from that submission.
Round 1
Reviewer 1 Report
Dear authors,
The study is well described, and we can easily understand the point of your work. You can find my comments and concerns in the text.

Reviewer 2 Report
Introduction describes lightly the theme and objectives of the present study. There is no description on the importance or action on the genes analysed for their expression. Missed information of the strain used as film-forming and some other comments that I described below.
The idea of co-culture and how they may affect the grown of F. graminearum and its possibilities for promote health on wheat by measuring FHB at greenhouse and field, observation by different microscopy, several proteins production, mycotoxin production and gene expression, is a well strategy and very creative one for the objectives of the present work. But somethings are missing in the manuscript and more details are needed to scientifically probe the information
Section 2.2: How many metabolites are you putting in the PDA media? 1:50 ratio may be 1ml of metabolite per 50 ml of media? How can you know if the metabolites are always the same amount and type in every repetition of the assay? Did you probe metabolite at field and green house experiment? (Take references).
Section 2.4: Elisa is not enough for Mycotoxin quantification, for many international regulations.
Section 2.6 Seed treatments with biocontrol microbial culture. There is no explanation on how talc power for coating is made up, and how can you probe that film formation is actually happening only by adding a film-forming agent. Please make a reference for this biofilm-forming agent and some description. Or, if you have other article with this information please cite.
Please make some reference on how the Relative disease severity % is calculated, there are several references old and new with a lot of discussion about this topic. Same comments on others calculation you made in the following parts of Materials and Methods.
2.7 gene expression; Why these genes? What information are they giving? References?? Is there any housekeeping gene to measure Fungal DNA on the samples? I think the gene names are known but if you are going to used them you need to explain and give a little more information about it.
Line 250, Point 3.4 is in a wrong place.
Line 181, there is a point following F. graminearum and inoculation
Results, picture and figures are quite remarkable but if you want to probe your objectives you need to have a background of knowledge of the theme and some more references on the know-how.
And discussion also need references!
